# Reconsidering the Nehushtan as a Magical Healing Device within the Geographical, Cultural, and Magico-Religious Context of the Ancient Near East

Gillian Williams and Mariette Harcombe *

Department of Biblical and Ancient Studies, University of South Africa, Pretoria 0002, South Africa;
malgill@sybaweb.co.za
* Correspondence: harcoam1@unisa.ac.za or amharcombe@gmail.com

**Abstract:** According to Numbers 21:4–9, the Nehushtan was a copper/bronze snake effigy that functioned as a 'magical' healing tool to cure the early Israelites from venomous snakebites they incurred during their desert wanderings. What is unclear from the narrative is the symbolic significance of the event, the materials used, the technical skills required, and whether magic was at play. Firstly, when considering the magical effects of the Nehushtan, we must define which type of magic—apotropaic or sympathetic—was involved. Based upon existing scholarship on the topic, the general consensus is that the Nehushtan represented sympathetic magic, underpinned in this instance by homeopathic/imitative magic. To highlight this point, this study will provide selected examples of both types of magic so that the Nehushtan's association with sympathetic magic can be illustrated. Secondly, and most importantly, we must consider why the image of a snake was chosen if the very affliction (envenomation) suffered by the people was caused by the creature now being posited as a symbol of divine healing. Did the ancient perceptions of snakes and healing play a role in this decision? Why did the early Israelites not question the logic behind the use of a magical snake effigy when both magic and effigies were technically prohibited by biblical laws? To answer these questions, the study will consider the historical background (the Exodus from Egypt), the set (geographical location), and the setting (cultural contact and influence) in which the narrative of the Nehushtan took place.

**Keywords:** Nehushtan; healing tool; Numbers 21:4-9; Ancient Near East; apotropaic and sympathetic magic; archaeology; snake; serpent

## 1. Introduction

According to the biblical narrative of Numbers 21:4–9, the Israelites grew weary of their seemingly endless journey through the wilderness towards the Promised Land. They complained about the lack of bread and water, and eating the same manna sent by God, day after day. Because of their ungrateful behavior, God sent fiery serpents (venomous snakes) against the people, and many were bitten and died. The people subsequently repented for their sins and begged Moses to appeal to God on their behalf, which he did. God then instructed Moses to make a fiery serpent and mount it on a pole/standard. All those who gazed upon it survived the ordeal.

What is unclear from the narrative is the symbolic significance of the event, the materials used to manufacture the snake image, the skills required to produce the effigy, and whether some form of magic—either apotropaic or sympathetic—was at play. The most important detail that is omitted, however, is whether existing traditions and exposure to similar cult objects from neighboring groups influenced the decision to construct a bronze snake effigy.

Therefore, one of the main questions that arise is whether existing cultural practices (including 'magic') and regional production technologies (i.e., mining, smelting, and forging)

influenced the construction of the Nehushtan through processes of enculturation. When considering which cultures the ancient Israelites were exposed to, one must remember that there is still no consensus among scholars as to (1) whether the Exodus was a real and singular historical event or (2) when the Exodus actually took place. Therefore, for the purposes of this study, we shall assume that the early Israelites who constructed the Nehushtan were on their way to the Promised Land, following their escape from Egypt, and that the construction of the Nehushtan occurred after the proposed timeframe for the Exodus: the 13th century BCE. The main purpose of this article is therefore to consider whether the early Israelites were influenced by cultural traditions found among the people(s) they encountered and to determine how/why it was acceptable for the early Israelites to resort to a magical cure from a snake effigy when both magic and effigies were technically prohibited by biblical laws (see Ex 20:3–6; Lev 19:26–28, 20:27; Num 22:17; 2 Kgs 21:6; Isa 47:9).

To answer these questions, the study will rely on existing research into the topics of apotropaic and sympathetic magic, snake symbolism and cult practices, and the healing properties of copper, among other topics. Prominent studies on the topic of magic and snakes and symbols of healing include those conducted by Golding (2013) on the perceptions of serpents in the Ancient Near East, Golding (2023) on snakebite cures in ancient Egypt, Amzallag (2009) on the connection between Yahweh and metallurgy, Ritner (1992, 2003) on magic in general and Egyptian magic in particular, Zinn (2012) on magic in Pharaonic Egypt, Retief and Cilliers (2005) on the connection between snake and staff symbolism and healing, and Antoniou et al. (2011) on the serpent rod as a healing symbol throughout history.

### 1.1. Introduction to Magic

According to Katharina Zinn (2012, p. 1), 'magic describes actions or knowledge relating to the supernatural or the demonic but not the divine'. Durkheim (1926) also distinguished between religion and magic, noting that religion works towards community goals, while magic is an exchange between a practitioner of magic and a client (cf. Liedeman 2002, p. 61). However, in the case of the Nehushtan, we see the application of 'divinely sanctioned' sympathetic magic within a religious framework towards the benefit of a community (the envenomated Israelites). The magic of the Nehushtan can be seen as 'divinely sanctioned', as it is not the snake effigy itself that provides a snakebite cure, but faith in Yahweh.

One should also consider that, for the greater part, there was no clear-cut distinction between good and evil magic. For example, according to Ritner (1992, pp. 189–92), magic was not perceived by the ancient Egyptians as a 'black' art or 'sorcery' that would have conflicted with 'white' or 'pure' religion. As magic was inherent in almost every aspect of Egyptian society, and considering the popular belief that the early Israelites had come from Egypt (i.e., the Exodus), Egyptian perceptions of magic as a 'neutral force' may still have persisted in the minds of the early Israelites during the Exodus. Considering the wider cultural context of the Ancient Near East, the early Israelites may also have been familiar with, and influenced by, the magic practices of other groups. But, before we consider examples of the latter, let us first review the two types of magic under consideration: apotropaic and sympathetic magic.

### 1.1.1. Apotropaic Magic

According to Coulson et al. (1980, p. 33), apotropaic magic is believed to have the power to avert bad luck or evil influences. The term is based on the Greek *apotropein* ('to turn away or revert'), *apotropaios* ('that which averts') (Faraone 1992 in Darby 2014, p. 6), and *apotrope* ('turning away or averting the evil eye') (cf. Shayne 2019).

Therefore, the underlying principle that governs apotropaic magic is the ability to avert or repel malevolent influences (i.e., curses, bad luck, poor health, etc.). Golding (2013, p. 25) notes that apotropaic magic boasts two characteristics: (1) the protective aspect

or tutelary, which is to repel harm or evil, and (2) the prophylactic or pre-emptive one against illness.

For apotropaic magic to exert its tutelary and/or prophylactic effects, certain actions need to be performed, or objects used. For example, apotropaic magic features active ritual observances, such as the performance of magical ceremonies or the casting of spells. It is also effective in a more passive role through the wearing of good luck charms, pendants, or amulets. Other 'superstitious' behaviors that are associated with apotropaic magic include acts like tossing salt over one's shoulder, knocking on wood, or crossing one's fingers.

According to researchers who study the psychology of human stress (see Keinan 1994, 2002; Mandal 2018; Lasikiewicz and Teo 2015), superstitious beliefs and their associated practices are more common during challenging or stressful situations. The main effect of using these objects is that they instill a sense of control in a situation where it is lacking. For example, it may grant the owner/wearer self-confidence when facing an adversary, or it may provide a sense of hope when fighting an illness or instill a sense of divine protection during pregnancy and childbirth. According to Damisch et al. (2010), superstitious behaviors may in fact improve performance through self-efficacy.

According to Hildburgh (1951, p. 235), amulets would bestow the wearer with the needed confidence to undertake a challenging task. For example, a breastfeeding mother could wear an amulet to increase her milk production. Here, Golding (2013, pp. 26–27) mentions that a mother's trust in the apotropaic function of an amulet could result in reduced tension and stress levels, which could in turn have a positive physiological effect on her breastfeeding schedule.

### 1.1.2. Sympathetic Magic

Sympathetic magic can take two forms: homeopathic (also referred to as imitative) magic and contagion magic. According to Frazer (1954, p. 11), it is the 'Law of Similars' that underpins the workings of homeopathic and imitative magic. According to this law, anyone who performs this type of magic relies on the premise that a desired effect can be manifested by imitating it (Wigington 2019). Joines (1974, p. 87) adds that sympathetic magic relies on the principle that the fate of an object or person can be governed by the manipulation of its exact image. For example, by stabbing a representation (either an image or a doll-like replica) of one's enemy, one can do harm or even destroy that individual. Another example would be 'rain dances' that are performed by multiple cultures across the globe. In these ritual performances, some aspects of rain (like the sound of thunder, or raindrops splashing on the ground) are imitated during the dance to manifest the desired phenomena.

In the case of contagion magic, the 'Law of Contact' is the driving force. Here, for the magic to be effective, the person/object who is the target or 'recipient' of the magic effect should have been in physical contact with the object, whether or not the object itself once formed part of the individual, like a lock of hair (Frazer 1954, p. 11). According to Rozin et al. (1989, p. 703), contagion magic relies on a transference of properties in which objects that were once in contact could influence one another. This connection between objects, and/or the people who interacted with them, remains in place even though their physical contact has been broken.

However, considering the basic underpinnings of contagion magic, it is safe to say that the Nehushtan is not an example of this type of magic, as the Israelites were not instructed to touch or physically interact with the snake effigy in any way to receive healing. All they needed to do was look upon it. Therefore, for the purposes of the paper, when referring to sympathetic magic, we are by default referring to its imitative/homeopathic form.

Considering the magical context of the Nehushtan, Liedeman (2002) acknowledges that the Nehushtan has 'a dimension of sympathetic magic', while Joines (1974, p. 87) asserts that sympathetic magic is a prominent element within the Nehushtan tradition, and not just a dimension thereof.

In the section that follows, selected examples of both apotropaic and sympathetic (imitative) magic will be presented, followed by the principles that underpin each example.

These principles will be used as far as possible as a theoretical framework to assist our reinterpretation of the Nehushtan narrative in Numbers 21.

## 2. Examples of Magic and How They May Relate to the Nehushtan

What is interesting to note is that superstitious behaviors, many of which relate to practices of both apotropaic and sympathetic magic, are believed to be culturally transmitted human behaviors (Mandal 2018, p. 65). Variations in the practices themselves can be explained through cultural evolution (social change over time and space) and environmental psychology (how set and setting impact our behaviors; Mandal 2018, p. 65). Because of the phenomena of enculturation, the following section will consider examples of both types of magic from the Ancient Near East, Egypt, and Greece.

### 2.1. Apotropaic Magic

As discussed above, apotropaic magic is the use of objects to ward off the forces of evil, or even to heal certain conditions. It involves spells and incantations (either spoken or written), the wearing of amulets, and the placement of objects in strategic localities to affect the desired magical outcome. These items can take various shapes and forms, including a talisman or jewelry inscribed with magic words (Posner 2007, p. 121), a bundle of roots placed within a cylinder and worn as a pendant Rosner (2000, p. 17), or even body parts belonging to one's ancestors or prominent figures (Schrire 1966, p. 5).

Apotropaic magic can be both preventative and curative. According to Burnett (2019, p. 73), 'amulets and amuletic plaques played an important role in apotropaic and healing contexts', as 'protection against illness, foreboding circumstances, malevolent magic, spirits, or deities was sought through magico-religious means'. In some instances, amulets are even believed to cure illnesses (Rosner 1977, 2000). Both applications (preventative and curative) have been in use since archaic times (Bar-Yosef Mayer and Porat 2008, p. 8548) and many have continued until this very day. According to Rosner (2000, p. 17), amulets were/are potent through the placebo effect, as the supposed effects of magic can be explained as resulting from the individual's expectations of the outcomes.

In the section that follows, we will consider a few examples of apotropaic magic from the Ancient Near East, Egypt, and Greece.

### 2.1.1. The Evil Eye

The evil eye is defined as 'a malicious or envious look superstitiously believed to do material harm' (The Oxford Dictionary), and Coulson et al. (1980, p. 289) add that it is the faculty, possessed by some, to inflict harm or injury through a malevolent look. Rosner (2000, p. 119) states that belief in the evil eye was common throughout much of antiquity. Berger (2013, p. 786) notes that its prevalence continued beyond the 7th century BCE, while Kotze (2013, p. 268) mentions that the evil eye can be dated to Sumerian times.

While the evil eye is not specifically mentioned in the Hebrew Bible, it is referenced quite often in the Babylonian Talmud (see for example Babylonian Talmud—Tractate Berachot 55b). While the evil eye refers to a malevolent look or stare that can inflict harm, evil eyes are also painted onto beads and amulets to counter the influence of the evil eye stare. A good example of the application of the evil eye in amulet form is the *hamsa* ('hand of God') amulet which is still popular among Jewish people today. Evil eyes as beads, painted stones, and a variety of different forms remain popular today as jewelry, keychains, wall decorations, and the like. The principle behind their magical working is 'what can harm, can also protect'.

### 2.1.2. Knotted Amulets

Day (1950, pp. 234–35) notes that knots possessed healing powers, with one example being described in a popular headache cure from the 9th century BCE. In addition, knots were also employed as part of a ritual to treat ophthalmia (inflammation of the eye), in

which black and white threads were twisted and knotted together while performing an incantation (Day 1950, p. 236).

From Jewish tradition, we see the use of *tallitot* (prayer shawls) featuring *tzitzit* (knotted tassels). According to Trepp (1980, p. 27), these knotted shawls date back to biblical times and are referenced in Numbers 15:38–40 as 'robes of responsibility'. The tassels serve as a reminder to their wearer to be faithful to God's laws and to resist their sinful desires. In a way, these tassels can be interpreted as having an apotropaic effect, as the sight of them deters the wearer from committing sinful deeds.

Bácskay (2019) also mentions the existence of Sumerian amulets known as 'head-stones' that consist of a 'head-stone' strung on 'seven knotted yarns made of wool' (Bácskay 2019, p. 186). Interestingly, the efficacy of the head-stone amulets relies on a magic analogous effect, as 'the name of the amulet stone includes the same logogram as the diseases' (Bácskay 2019, p. 155).

In ancient Egypt, the tying and untying of knots played a crucial role in magic, and knots are often presented on amulets, or incorporated in the string used to suspend an amulet. One example is the *tyet* amulet, which was a popular protective amulet worn by pregnant women. The amulet is knot-shaped, and although the meaning behind the shape is unclear, it is believed to resemble the knotted girdle of the goddess Isis (Gahlin 2010, pp. 204–5).

Thus stated, knots can be used (indirectly) to prevent the wearer from committing sinful acts, while they can also serve apotropaic healing functions through imitative magic.

### 2.1.3. Medicinal Amulets

Medicinal amulets that feature apotropaic incantations are well known throughout ancient times. Some of our best examples include the amulets of Arsinoë. These thin sheets of metal (one made in silver and the other gold) date from the first century CE and are engraved Aramaic incantations that protect the owner—a Greek lady, Arsinoë—from a broad spectrum of malevolent forces and illnesses (Kotansky 1991, p. 267). These thin metal amulets were also used by the Egyptians and Phoenicians from the fifth to seventh centuries CE (Kotansky 1991, p. 268).

Several Neo-Assyrian and Late Babylonian medical compendium tablets feature a variety of incantations against common ailments, including headaches and migraines (Bácskay 2019). We also find examples of inscribed cylinder-shaped beads from the tomb of Assyrian queens (one possibly being Ataliya, queen of the Assyrian king Sargon II). The incantations inscribed on the beads served to ward off headaches/migraines, and the beads were worn as part of a necklace (Bácskay 2019, p. 175). According to Liedeman (2002, p. 46), the Israelites wore charms to protect them from snakebites. What is interesting to note is that 'the application of medical incantations written on amulet stones represents the practical use of these amulets preserved in medical compendium tablets' (Bácskay 2019, p. 187). In short, inscribed necklaces served almost like a portable 'extension' of the incantations presented in medical compendia, like how one would use a 'pocket guide' version of a publication in modern times.

### 2.1.4. Egyptian Snakes and the Uraeus

In ancient Egypt, magic was a regular component of religion (Taylor 2001, p. 186), and there was no conflict between magic (as 'black arts' or sorcery) and religion (Ritner 1992, p. 189). The words for amulet included *sa*, *meket*, and *nehet*, all of which derived from verbs that mean 'to guard' and 'to protect'. Another word associated with amulets was *wedja*, which translates as 'well-being' (Gahlin 2010, p. 204).

Snakes and snake symbolism played an important role in both apotropaic and sympathetic magic in Egypt. This can be seen in the examples of the *uraeus* and the burning of wax effigies of Apophis. The *uraeus* was regarded as the symbol of ultimate power in Egypt, and people saw the *uraeus* as they looked up to the pharaoh (as he was usually physically

raised above the populace). In this instance, the snake image (*uraeus*) functioned to protect against or to ward off evil.

What is interesting to note is that a common tactic for repelling snakes was to employ snake imagery in apotropaic contexts. The *uraeus* was sometimes referred to as *Weret Hekau* (Gahlin 2010, p. 191), with the latter being a cobra-shaped goddess whose name translates as 'Great of Magic' (Zinn 2012, p. 4). According to Ritner (2003, p. 193), *heka* was a 'morally neutral' and 'divinely sanctioned' magical force that existed in all things throughout the universe (cf, Golding 2013, pp. 18–19). Thus, it can be argued that objects like the *uraeus* were 'powered' by this magical force.

Shanks (2007) is of the opinion that the Nehushtan operated along similar lines as a *uraeus* guarding mummies from the serpents of the underworld. It is also possible that the Nehushtan's healing powers were similarly produced through a type of divinely sanctioned magical force (cf. Ritner 2003) resembling the Egyptian *heka*. In this light, a snake amulet or effigy can be used to protect against or to ward off evil, with the efficacy thereof being linked to the principle of 'like repels like'.

Snakes were also associated with healing in indirect contexts. For example, the goddess Serqet (Selkis) was mostly depicted as a scorpion goddess but was also referenced as a snake goddess (Watterson 1996, p. 85). The priests of Selqet specialized in treating snakebites and were often employed in situations (like mining expeditions) where the risk of snakebite was high (Ritner 2003, p. 195; Golding 2013, 2023).

The connection between snakes, copper, magic, and healing will be discussed in greater detail later in this article.

### 2.2. Sympathetic Magic

The Collins Dictionary defines sympathetic magic as 'magic in which it is sought to produce a large-scale effect, often at a distance, by performing some small-scale ceremony resembling it, such as the pouring of water on an altar to induce rainfall'. Frazer (1954, p. 10) describes this principle as 'like produces like' or 'an effect resembles its cause'. From the magician's point of view, he or she can influence an effect by imitating its cause. This category of sympathetic magic is also termed the 'Law of Similars', otherwise known as homeopathic or imitative magic.

Frazer (1954, p. 12) defines homeopathic/imitative magic as an attempt to destroy, injure, or inflict harm on one's enemy by damaging or annihilating an image of that person. The image can take many forms: a drawing in the sand, an effigy painted on a stuffed straw doll or a log of wood (Frazer 1954, p. 12), an old sock stuffed with sawdust, or twigs shaped to resemble a human body (Weston 2015); in fact, anything intended to represent the foe can be used. But, in addition to malicious intent, imitative magic also has positive uses and applications that relate to everyday life, such as food harvesting. In this regard, Frazer (1954, p. 12) stipulates that the maker's intention is the critical element of such a ritual.

Frazer (1954, p. 11) believes the practice of imitative magic is thousands of years old and was well known in ancient India, Babylon, Egypt, and Greece. Interestingly, this type of magic is still in use today in places such as West Africa, North America, and Canada, to name but a few.

The following examples of sympathetic magic throughout the region add to a better understanding of its practice in the ancient world:

### 2.2.1. Tutankhamun's Ceremonial Sandals

A good example of sympathetic magic is the ceremonial sandals of King Tutankhamun. Hawass (2013, p. 139) explains that these sandals were found in the tomb, and are made of wood and inlaid with leather, gold, and tree bark. The upper surface of each sandal is decorated with images of a Nubian and an Asiatic prisoner. Their arms are tied behind their backs, and a rope can be seen around the necks of both men. These men represent the traditional and sworn enemies of Egypt, who must be reminded of their enduring capitulation to the pharaoh. This is achieved by the ruler wearing and thus figuratively

walking on their supine figures. Here, we see an attempt at destroying or harming one's enemies by damaging an image of them. This illustrates the imitative magic principle of 'like kills like', 'an effect resembles its cause', and the 'Law of Similars'.

### 2.2.2. Theriacs in Greece

Theriacs are medical concoctions that rely on the principles of homeopathy and can be traced back to antiquity (Parojcic et al. 2003). Popular in ancient Greece, these potions made of herbs, spices, honey, and animal parts remained popular until the 16th century CE (Karaberopoulos et al. 2012). According to Coulson et al. (1980, p. 882), theriacs often served as antidotes to poisons and snakebites, and Pliny the Elder (23–79 CE) mentioned that they were particularly useful against envenomation (Karaberopoulos et al. 2012).

Interestingly, the Roman physician Andromachus (54–68 CE) modified an age-old theriac by substituting minced lizard (Dioscorides et al. 2000, p. 70) with viper meat (Dioscorides et al. 2000, p. 18). According to Parojcic et al. (2003), Andromachus probably considered the 'Law of Similars' (in which 'like cures like' and 'what harms can also heal') when making the substitution.

Like ingesting an acidic substance to cure heartburn (Williams 2009, p. 338; 2023), or consuming the heart of a lion to obtain its strength and bravery (Barrett 2002), the ingestion of viper meat could grant one the power to resist their venomous effects, as snakes are immune to their own venom (Eggleston 2018, p. 1). These beliefs appear to have some foundation in reality, as Golding (2013, p. 167) and Holland (2013, p. 70) observe that modern scientists have studied different snake venoms as treatments for cancer and autoimmune disorders, as well as for pain relief.

### 2.2.3. Sympathetic Magic in the Hebrew Bible

The biblical account of Jacob's speckled and spotted sheep (Gn 30:37–31:16) is a good example of sympathetic magic. To breed many striped, speckled, and spotted animals, Jacob peels the bark from fresh shoots of poplar, almon, and plane trees and sets the staves up where the animals can see them while drinking water or mating. Not only were the resulting lambs and kids striped, speckled, and spotted, but the ones that Jacob wanted were much sturdier than the rest.

According to scholars like Plaut (2006, p. 210) and Westermann (1981 in Noegel 1997, p. 7), a common notion existed that visual stimuli received at the moment of conception may influence the characteristics or appearance of the offspring. This is a prime example of the 'Law of Similars' in which 'like produces like'. Perhaps, as the sheep looked at the colored staves and somehow 'absorbed' their characteristics, gazing upon the Nehushtan would also imbue the observer with the snake's immunity against its own venom.

Another example of sympathetic magic is the sweetening of the bitter waters of Marah (Ex 15:22–25). When Moses and the Israelites came to a place called Marah, they found that the water there was bitter and undrinkable. God then instructed Moses to throw a branch of bitter wood into the water to sweeten it. In this case, the bitter waters were cured when a branch of bitter wood was thrown into the water. This type of healing magic is another example of homeopathic or imitative magic where 'like cures like' and 'what kills can cure'.

The story of the Philistines, the golden mice, and the tumors (1 Samuel 6) presents a similar example. In this narrative, the Philistines defeated the early Israelites at the battle of Eben-ezer and took the Ark of the Covenant as booty. The next day, the Philistines found the statue of their god Dagon, which they had left standing next to the Ark, toppled and decapitated. Ignoring this warning, they kept the Ark, only to awaken the next day covered in boils/tumors and pestered by a plague of mice/rats (cf. Firth 2009).

In their response to this crisis, the Philistines created models of the tumors, which 'treated and cured' the pestilence, as well as golden effigies of rats, which they dispatched in a cart to the Hebrew God as compensation for having taken the Ark of the Covenant (1 Sm 6:3–4) (cf. Firth 2009, p. 102). It is possible that the Philistines made a connection between the rats and the boils, attempting to control the situation through sacrifice and

sympathetic magic (Langdon-Brown 1941, p. 34). By sacrificing the cows that had taken the Philistine cart back into Israelite territory, they were trying to produce the desired effect (the death of the rats) by imitating it.

## 3. The Roles of Snakes and Copper in Ancient Healing

### 3.1. Snakes, Healing, and Serpentine Rods

According to Retief and Cilliers (2005, p. 553), an ancient Egyptian hymn to the goddess Merseger recalls the story of a man from Thebes who was healed from his afflictions after the goddess visited him in snake form. The Sumerian god Ningishzida, who was the son of the healing god Ninazu, is depicted with a staff featuring two entwined serpents (Retief and Cilliers 2005, p. 553). Similarly, we encounter the Rod of Asklepios, around which one snake is coiled, and the Emblem of Hermes, which is encircled by two snakes (Retief and Cilliers 2005, pp. 554–55). The snake-coiled staff continued to serve as a healing symbol throughout much of history and was revived during the Renaissance to identify apothecaries that stocked antidotes to snake venom (Antoniou et al. 2011, p. 218; Prakash and Carlton 2015).

Although there is a widespread association between snakes and healing in the ancient world, we now focus our attention on the cult of Asklepios. Antoniou et al. (2011, p. 219) note that the parallels between the Nehushtan and the Rod of Asklepios are clear: 'healing by the power of God through a symbolic snake upon a staff'.

The Greek god of healing, Asklepios, is described by Homer in the Iliad (IV, 194) as a 'blameless physician' (Salem and Salem 2000, p. 169). Healing centers (known as *Asklepieia/Asklepions*) associated with the god were known from as early as the late 6th century BCE (Lock et al. 2001, p. 710). Examples of such therapeutic retreats include those located at Epidavros, Athens, Kos, and Corinth (Leonard 2019).

Patients—who in most instances had exhausted all other attempts at receiving cures for their ailments—could spend the night at an Asklepion in the Abaton and have their dreams interpreted by the priests, who were well versed in medicine, the next morning (Charitonidou 1978, pp. 10–15). Situated in remote locations, like mountainous valleys, *Asklepions* often featured mineral springs or sources of running water, which in themselves were reputed to have healing powers (Charitonidou 1978). However, the most interesting elements are the stories that recall the presence of healing snakes that would freely roam the grounds of these sanctuaries, visiting patients and even 'licking' wounds or afflicted areas (Downing 1990). It has been proposed that the snakes in question were non-venomous constrictors, possibly the Aesculapian rat snake (*Zamenis longissimus*) (Böhme and Koppetsch 2021, p. 481).

Edelstein and Edelstein (1975) recall the story of an unnamed man who visited an Asklepion to receive healing for his afflicted toe. According to the story (*Inscriptiones Graecae* iv, I, pp. 121–22), the man received healing after a snake visited him and he dreamed that a handsome man had applied ointment to the diseased area (Wells 1993). In another example, a woman known as Agamede of Keos, who was unable to fall pregnant, visited an *Asklepion*. During her visit, a snake lay on her stomach—perhaps representing the *chthonic* (earth-related) form of the god Asklepios—whereafter she managed to conceive and birth five healthy children (Edelstein and Edelstein 1975, p. 237). Snakes could also assist with ailments such as consumption (pulmonary tuberculosis), as was the case with Thersandrus of Halieis, who was healed following a visitation from a snake (Edelstein and Edelstein 1975, pp. 235–36).

References to snakes 'licking' wounds at *Asklepions* are interesting, as snakes are not known to lick their prey. However, there is a possible medical explanation for the connection between snake saliva and healing. According to recent studies, the saliva of certain snakes contains polypeptide amino acids and epidermal growth factors that stimulate healing (Angeletti et al. 1992, p. 224; Williams 2009, p. 60; 2023).

When considering the similarities between the Rod of Asclepius and the Nehushtan, the act of looking at or gazing upon the snake effigy is intriguing. According to Ovid

(*Metamorphosis, Book 15*), Asklepios said: 'Only *look upon* the serpent [emphasis added] that twines about my staff, I shall change myself into this [emphasis added]', and he transformed himself into a snake (Charlesworth 2010, p. 255). This instruction resembles the next of Numbers 21, verse 9, where the Lord said to Moses 'Make a snake and put it on a pole; anyone who is bitten can *look at it* [emphasis added] and live.' As emphasized, the act of 'looking upon' the snake effigy closely resembles the story of the Nehushtan, as the early Israelites simply had to look to the Nehushtan to receive healing.

### 3.2. Copper and Healing

The ancient association between copper and healing is also noteworthy. For example, the Edwin Smith Papyrus notes that battle wounds can be dressed with copper pieces to avoid infection (Grass et al. 2011; Keevil 2017). The connection between copper and healing has been validated by modern research, as studies have shown that copper exhibits broad biocidal properties that may prevent infection and promote healing (Borkow et al. 2010).

What is also compelling is the direct connection between snakes and copper, in that snakes were believed to absorb minerals (like copper) from the earth (Borkow et al. 2010; Grass et al. 2011; Keevil 2017). This would imply that snakes possessed healing qualities because they absorbed copper.

A snake like *Echis coloratus*, which has a striking copper-colored appearance and inhabited the copper mining areas of Arabah and Sinai (Amzallag 2015, p. 99), would therefore have possessed definite healing properties according to regional inhabitants. It can therefore be argued that the existing symbolic and practical (biochemical) associations between copper, healing, and snakes were clear enough for the ancient Israelites to not question the logic behind the production of a copper/bronze snake effigy to heal snakebites.

## 4. Discussion

### 4.1. A Recap of the Biblical Narrative

During their journey to the Promised Land, en route from Mount Hor via the Red Sea (Num 21:4), the early Israelites became dissatisfied with the lack of bread and water, and their boring diet of manna. Some even questioned why they had left Egypt just to die in the wilderness (Num 21:5). God punished them by sending venomous snakes among the people. Many were bitten and died (Num 21:6). Traumatized by this ordeal, the people repented and pleaded with Moses to appeal to God on their behalf, which he did (Num 21:7). God then instructed Moses to make a fiery serpent and set it on a pole, as those who look at it will live (Num 21:8). Moses then made a snake of bronze and placed it on a pole. Anyone who looked at the Nehushtan survived the snakebite tragedy (Num 21:9).

When considering this narrative, a modern reader might find themselves slightly perplexed as to why God would instruct Moses to produce a snake made of bronze and set it atop a pole, considering the prohibition(s) against effigies and magic. Why could God not simply heal those inflicted by venomous snakebites through direct miracles? The answer to this question may be found by examining the historical background of the early Israelites (i.e., their Exodus from Egypt), the set (geographical area), and the setting (cultural contact and influence).

### 4.2. Historical Background

Firstly, let us consider the events that brought the early Israelites to wander through the wilderness. It must be noted that while the biblical account of Exodus tells the story of an enslaved people escaping the tyranny of Egypt, scholars still disagree on whether the Exodus actually occurred as a singular historical event (or even at all). Yet, for the purpose of this investigation, let us imagine that the early Israelites did indeed escape from Egypt after many centuries of enslavement.

This would mean that the early Israelites were strongly influenced by the mythology and cultural practices of ancient Egypt—including an understanding of what magic is, how snakes possess a certain duality (as both threat and protector), and how copper possesses

healing characteristics. With this knowledge in mind, the early Israelites would not have questioned the efficacy of employing a copper snake effigy in healing. When considering the material best suited for the intended purpose, copper or bronze would have made perfect sense, as the connection between copper/bronze and healing was already well established within their 'mythological frame of reference'. But most importantly, the people had faith in this strategy as it came from the Almighty.

*4.3. Set (Geographical Location, Mining Resources, and Skills)*

Secondly, let us consider the set (geographical location) of the early Israelites at the estimated time of the snake plague. The Bible mentions that they were en route from Mount Hor via the Red Sea so as to avoid Edom. Their journey would have brought them into contact with the local inhabitants of the area, including the Kenites who were well-known metallurgists.

The Bible tells us that Moses made a bronze serpent, but was it really Moses himself who made it? It is unlikely that Moses made the bronze effigy himself, as the entire process from start to finish would have required the mining of natural resources, smelting, and casting the serpent. Although these skills and knowledge sets could have been present among the group, they could have sought the assistance of local groups.

It is also possible that the early Israelites commissioned the Kenites, who were itinerant metallurgists and lived in the vicinity of copper mines, to manufacture the Nehushtan (Sperling 2008, p. 76). As there is no textual evidence to support this claim, one should look to the archaeological record to consider its probability.

What is compelling is that snake effigies have been discovered at several cultic locations, including Megiddo, Jebus, and Timna. According to Milgrom (1990, p. 173), an Egyptian copper mining and smelting site, dated from 1200–900 BCE, was located near Punon (modern-day Feinan) (cf. Rothenberg 1990a, p. 8). We also know of other copper industries in Timna (western Arabah) and Feinan (eastern Arabah) that existed since the 4th millennium BCE (see Rothenberg 1990a; King and Stager 2001; Hauptmann et al. 1999). Archaeological remains of a casting workshop and copper snakes at the Egyptian temple of Hathor at Timna also provide insight. The location was first mined during the time of Seti I (1323–1279 BCE) or Rameses II (1303–1213 BCE), and Rothenberg (2007, p. 727) believes that the shrine to Hathor was erected during the proposed time of the Exodus. Rothenberg (2007, p. 727) also notes that Timna was occupied by the Egyptians, the Midianite/Kenites, the Amalekites from the Negev, and the Midianites from Hijaz (North-western Arabia) (Rothenberg 2003, p. 9).

According to Dunn (2014, p. 388), the Midianites took control of the mines during the 14th to 12th centuries BCE, and Avner (2014, p. 1105) adds that they constructed a tented shrine (Avner 2014, p. 1105). Rothenberg (2007, p. 727) adds that the variety of objects recovered from the site provides a glimpse into the relationship between the Israelites, the Kenites, and the Midianites. A 12 cm long bronze snake, dated to 1300–1100 BCE, was discovered in the ruins of the temple of Hathor (Münnich 2008, p. 41), and Rothenberg (1990b, p. 175) believes that it could be contemporaneous with the biblical Nehushtan.

Another point to consider is the presence of venomous snakes in the area. The term 'fiery serpent' is often used when referring to both the snakes that attacked the early Israelites (Num 21:6) and the Nehushtan itself (Num 21:8). In this regard, there are clear associations between the color of fire and that of copper, bronze, and the snakes that inhabit the area.

According to Rabbi Assi (in the *Midrashim*), the word 'copper' is similar to the word 'snake' (*Midrash Genesis Rabbah* 31:8). It is therefore possible that Moses was guided by the substantive name of the metal (*nekhoshet*), with copper being like the Hebrew word *nahash*, or snake. In addition, there is a connection between *śārāph* ('snake') and *śerāphîm* ('serpents' or 'venomous snakes'), which according to Rodríguez (2011, p. 1) could indicate that the snakes referred to as *śerāphîm* in Numbers 21 (verses 6 and 8) were capable of inflicting bites that result in burning sensations and severe inflammation. To this, Amzallag

(2015, p. 99) adds that the word *śārāph* is associated with a species of snake, *Echis coloratus*, which is native to the mining areas of Arabah and Sinai. The species is known for its dark orange-brown skin that resembles the color of copper. Another interesting fact is that one of the wadis in the Timna Valley is known as the *Nahal* Nehushtan (Beyth et al. 2013, p. 4). These facts clearly illustrate the connection between copper, the snakes that inhabited the geographical area, and the fiery pain they inflict through their venomous bites.

*4.4. Setting (Cultural Contact and Influence)*

When we consider the 'setting', we remind ourselves that long-term interaction between Greece, Egypt, Sinai, and the whole of Canaan gave rise to various expressions of hybridization and multiculturalism in the early Israelite religion and culture. They would have been exposed to, and familiar with, the cultural practices of the Ancient Near East.

This is relevant when considering the production of magical objects and the performance of spells and incantations. Whether the Bible explicitly mentions it or not, the possibility exists that some type of imitative magical rite was performed by the early Israelites, which possibly involved the metallurgists who produced the Nehushtan. What should be kept in mind here is that ancient metallurgists themselves were seen as 'magicians' in their own right (Amzallag 2009, p. 7), so their very involvement in the Nehushtan's production instills magic as an element of production by default.

It is also possible that the Nehushtan may have been influenced by the existence of snake cults in Canaan and early Israel that employed apotropaic and sympathetic magic. If a group like the Kenites was involved in manufacturing the Nehushtan, the early Israelites would also have been in contact with the cultural practices of that group. When considering the bronze snake effigies that have been found at locations like Megiddo, Tel Mevorakh, Shechem, and Gezer (i.e., Canaanite city-states under Egyptian rule and influence), Münnich (2008, p. 48) suggests that these effigies were venerated at local shrines, as is evident from the shrine of Hathor. Münnich (2008, pp. 48–49) also draws parallels between the Canaanite god Horon, the Phoenician god Eshmun, the Greek doctor Asklepios, and the Nehushtan, as these entities all represent a connection between snakes and healing. To this, Handy (1995, p. 41) adds that it is possible that the concept of the Nehushtan as a healing serpent could be linked to a heretofore unknown/unidentified god of snakebite cures.

When considering the possibilities of cultural contact, it is highly probable that the early Israelites would have been familiar with the workings of both apotropaic and sympathetic magic. By observing the effects of magic through cultural context, the early Israelites would not have questioned the 'logic' behind the construction of a copper snake effigy and would have readily believed in its ability to heal. In addition, they lived a life of suffering, and enduring a snake plague and living to tell the tale would have restored their fortitude as they proceeded on their journey towards the Promised Land. This falls within Collier's (2014) reference to the psychological forces at play when considering the efficacy of magical beliefs.

The Nehushtan was an exception to the rule when it came to magic practices, but not uncommon in their Ancient Near Eastern context. The Nehushtan functioned as sympathetic magic (benevolent) and not contagion (sorcery/witchcraft) magic, which means that the rules could be bent in terms of the use of magic. The early Israelites still knew that only YHWH could heal or rescue them (see the context of the Old Testament, e.g., Jdg 2:1-3). Antoniou et al. (2011) note that we encounter a reference to the Nehushtan in New Testament: 'just as Moses lifted up the serpent in the wilderness, so must the Son of Man be lifted up, that whoever believes in him may have eternal life' (John 3:14–15) and add that we encounter snake symbolism in religious iconography throughout Christian history.

**5. Conclusions**

Although magic and effigies were (generally) prohibited by biblical laws (see Ex 20:3–6; Lev 19:26–28, 20:27; Num 22:17; 2 Kgs 21:6; Isa 47:9), this rule was not strictly enforced

if a serious need for a 'magical' solution arose. As the Nehushtan served a sympathetic, benevolent function, and not contagion magic (more akin to sorcery and witchcraft), this healing snake effigy appears to have been an exception to the rule prohibiting (or at least restricting) magic. After considering examples of both apotropaic and imitative magic, it may be seen that imitative magic is applicable to the Nehushtan, rather than apotropaic magic.

When considering their historical background (their enslavement in Egypt and subsequent Exodus), the set (geographical location and availability of both raw materials and metallurgical experts), and the setting (cultural contact and influence), the early Israelites would not have questioned the logic or efficacy of constructing a snake effigy for the purpose of healing snakebites. The various examples of correspondence—between snakes and serpents, the color and heat of fire, the burning sensation of snakebites, the physical color of snakes (like that of *Echis coloratus*), the color of copper/bronze, the medicinal properties of copper, copper's resistance to corrosion, the supposed supernatural characteristics of snakes (shedding = rejuvenation, awaking from brumation = resurrection), and the medicinal properties of snake meat/venom (in theriacs)—make the association between snakes and divine healing appear almost logical.

Furthermore, it is possible that the mechanics of sympathetic magic can be explained by human psychology. If the early Israelites' belief in the curative abilities of a sympathetic healing device like the Nehushtan was already well established through their observation of magical cause and effect across generations, then the healing experience was actuated through self-efficacy. They believed it could work because it worked in the past, and this belief effected a type of 'magical placebo effect'. It can therefore be argued that a psychological force, rather than a true magical force, was at play. In conclusion, when considering the narrative of the Nehushtan, this study did not aim to uncover the hidden symbolic significance of the event, but rather focused on the cultural practicality and psychological underpinnings (so to speak) that make the event more plausible. The study therefore argues that the existing symbolic and practical (biochemical) associations between copper, healing, and snakes were clear enough for the early Israelites not to question the logic behind the production of a copper/bronze snake effigy to heal snakebites.

**Author Contributions:** Conceptualization and methodology, G.W.; supervision and writing, M.H. All authors have read and agreed to the published version of the manuscript.

**Funding:** University of South Africa (Doctoral Bursary). This research received no external funding. The APC was funded by the University of South Africa.

**Institutional Review Board Statement:** Not applicable.

**Informed Consent Statement:** Not applicable.

**Data Availability Statement:** Data are contained within the article.

**Acknowledgments:** We would like to thank the University of South Africa for the doctoral bursary provided to Williams towards the completion of her degree. We would also like to thank Magdel le Roux for her continued support and academic mentorship.

**Conflicts of Interest:** The authors declare no conflict of interest.

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
