# Peer review of "Reconsidering the Nehushtan as a Magical Healing Device within the Geographical, Cultural, and Magico-Religious Context of the Ancient Near East"

_religions, doi:10.3390/rel14111404_

Round 1

Reviewer 1 Report

Comments and Suggestions for Authors

Plus:

- very interesting topic, worthy of academic investigation

- good approach (general intoduction, explanation of the broader context...)

Minus:

- very general (shallow) information in the intro, data taken from basic (encyclopaedias), old (Frazer) or rather popularising literature (eg. Gahlin)

- information from crucial academic sources are missing

- for example, works of Bohak and Ritner cannot be omitted from papers on Jewish (Bohak) and Egyptian (Ritner) magic

- the author concentrates too much on the intro (types of magic), so the main topic (the Nehushtan) remains uninvestigated.

- there are possible direct parallels of the Nehushtan in Egypt and the Hellenistic world, but the author does not mention them

I suggest:

- to use better academic sources even for the intro

- to search for direct parallels of the Nehushtan in the ancient world (erected snakes on poles)

- to focus more on the main topic than on the intro

- rewrite the article

Comments on the Quality of English Language

- no major comments

- nice language, well readable

- only minor errors

Author Response

  1. The introduction has been improved
  2. Rotner 1992 and 2003 have been included.
  3. The authors feel that the topic of te Nehushtan has indeed by investigated by means of a comparative study between it and similar examples of snake healing cults and magic where snakes are involved in healing.
  4. I discussion on the cult of Asklepios has been added.
  5. A number of academic sources have been added, including Barret (2002) Bohme & Koppetsch (2010), Borkow, Orkon-Levy and Gabbay (2010), Darby (2014), Durkheim (1926), Golding (2023), Grass, Rensing & Solioz (2011), Keevil (2017), Kotansky (1991), Retief & Cillers (2005), Ritner (1992; 2003), Shanks (2007), Shayne (2019), Taylor (2001) Williams (2023), Zinn (2012).

Reviewer 2 Report

Comments and Suggestions for Authors

The style is somewhat prolix and occasionally repetitive, but the point is made clearly.  The references to archaeology are good, though they hardly require the story to be from the 13th century BCE--it could be centuries later.  Lines 102-6 look like a holdover from use of the material in a teaching context, and lines 414-15 are instructions to editors: both need to be edited out.  The word "behavior/behaviour" strangely appears both in its American and British spelling.  The Egyptian god is usually given as "Seth" rather than "Set" (line 302).  Typos on lines 308, 467, and 500.

Comments on the Quality of English Language

Give above.

Author Response

  1. The language style has been improved.
  2. The holdover text in lines 102–106 have been removed, as well as the instructions to authors in lines 102–106.
  3. American spelling has been applied across the text.
  4. The name Set has been replaced with Seth.
  5. The typos on lines 308, 467, and 500 were corrected.

Reviewer 3 Report

Comments and Suggestions for Authors

Overall this is an interesting and informative article and I enjoyed reading it.

1) I have marked the first point above as 'can be improved' for the simple reason that I think a brief literary review may enhance this article. Who else has written about Nehushtan, or what angle do they take? A summary of other scholars view points is needed in order highlight why your viewpoint is different. Have other scholars considered your question to the same extent, if at all? If there is nothing similar then it is perfectly fine to point this out. This only adds to highlight out why your work is important. 

2) Other minor points are included in the attachment for you to download. No major corrections. I am interested to know why copper was believed to have supernatural characteristics. It is worth including a bit more information on this. Copper may have had certain desired qualities and these qualities may have affected the reason why it was chosen as a material for the making of Nehushtan. 

Author Response

  1. The authors have added a brief literary review of the most crucial sources.
  2. The authors believe that their justification behind the study is supported by their identification of the gaps in the existing knowledge base. Considering why the early Israelites so readily accepted the snake as a healing symbol (because of enculturation) has not been covered by many authors, as most have focused on the type of magic employed.
  3. The authors have added information on the healing properties of copper and why the ancients believed that copper had supernatural healing powers (these powers were not really supernatural but had to do with the anti-microbial properties of copper).

Round 2

Reviewer 1 Report

Comments and Suggestions for Authors

The author has revised their work well. I agree with publishing. Yet, the title is still a bit misleading, as the nehushtan is dealt with only as a minor (but still interesting) case-study here.